# Foley Catheter as a Tourniquet for Hemorrhage Prevention during Peripartum Hysterectomy in Patients with Placenta Accreta Spectrum (PAS)—A Hospital-Based Study

**DOI:** 10.3390/life13081774

**Published:** 2023-08-19

**Authors:** Jakub Staniczek, Maisa Manasar-Dyrbuś, Ewa Winkowska, Kaja Skowronek, Rafał Stojko

**Affiliations:** Chair and Department of Gynecology, Obstetrics and Gynecological Oncology, Medical University of Silesia, 40-211 Katowice, Poland; jstaniczek@sum.edu.pl (J.S.);

**Keywords:** placenta accreta spectrum, placenta, postpartum hemorrhage, Foley catheter

## Abstract

Background: Placenta accreta spectrum (PAS) is a clinical entity significantly increasing the risk of a peripartum hemorrhage. Various surgical methods have been described in the literature, which aim to reduce the risk of bleeding, although they often lack reproducibility and have been performed on low numbers of patients. The aim of this study was to evaluate the use of the Foley catheter as a cervical tourniquet during cesarean sections, in patients with PAS. Methods: All patients who underwent peripartum hysterectomy due to PAS in a large single-center registry were included in the present analysis. The general demographics and clinical characteristics of all participants, including blood loss, and maternal and fetal outcomes, were collected and analyzed. Results: Twelve participants were included. The mean blood loss was 1200 ± 760 ml during operation and the mean ± SD procedural duration was 89 ± 17 min. The median (Q1–Q3) length of hospital stay post-procedurally was 5 (4–6) days. None of the patients required subsequent urgent surgical procedures after hysterectomy. The median (Q1–Q3) packed red blood cell units transfused in our cohort was 2 (0–3). Conclusion: Using the Foley catheter as a tourniquet might be an effective method of excessive bleeding prevention in patients with PAS during peripartum hysterectomy.

## 1. Introduction

The placenta accreta spectrum (PAS) encompasses conditions where the placenta directly adheres to the uterine myometrium. It is further subclassified as placenta accreta, increta, and percreta based on the depth of myometrial invasion. This clinical entity is commonly diagnosed. Advanced prenatal imaging methods, such as comprehensive obstetric ultrasound and magnetic resonance imaging, combined with a better understanding of the typical progression of PAS, have contributed to a rise in its prenatal diagnosis. Early identification of PAS is crucial, as it reduces adverse maternal and fetal outcomes. Notably, pooled data suggest risks including a 52% chance of peripartum hysterectomy, 47% risk of severe hemorrhage requiring transfusion, and some risk of maternal death due to PAS [1]. The potential for significant blood loss, possibly up to 7000 mL, should always be considered [2]. With the right prenatal diagnostic strategies, specialized medical centers can prepare tailored treatment plans [3].

Peripartum hemorrhage is a serious concern in obstetrics. In extreme cases, it mandates a peripartum hysterectomy, crucial for saving the mother’s life. Given the global burden of maternal morbidity and mortality, it is essential to continually explore innovative techniques to manage hemorrhage during caesarean section (CS). Rapid cessation of uterine bleeding is imperative, leading to the development of various rapid vessel ligation methods. The perioperative mortality from peripartum hysterectomy can be as high as 1.0%, with over 10% of patients sustaining injuries to adjacent organs, most commonly the urinary bladder [4]. Thus, effectively preventing and managing hemorrhage during peripartum hysterectomy remains challenging.

PAS arises due to a defect in the boundary between the endometrium and myometrium, preventing the uterine scar region from undergoing normal decidualization. This allows the anchoring placental villi to penetrate more deeply, increasing myometrial adhesion [5].

With the rising number of prior cesarean sections, the incidence of PAS is also growing, a trend expected to continue [6,7]. The presence of PAS poses significant risks including hysterectomy, intrapartum and postpartum hemorrhage, and maternal morbidity and mortality [8,9,10]. Thus, managing patients with PAS and peripartum hemorrhage becomes even more intricate.

Traditional hemostatic measures, such as uterine compression sutures and arterial embolization, are promising but can be technically challenging or bring potential complications. Other methods, though documented, might lack reproducibility, especially in complex anatomical situations, may not be as effective, or could necessitate additional resources, elevating costs [11,12,13]. A more straightforward, cost-effective, and accessible technique is needed, even for less experienced clinicians. The Foley catheter, primarily used for bladder drainage, has shown potential in various surgical specialties. Its ability to exert controlled pressure on vessels, temporarily halting blood flow, makes it a strong contender for managing hemorrhage during CS in PAS patients. While some sources discuss temporary clamping or ligation of uterine vessels or the internal iliac artery with surgical tools, these techniques are technically more demanding compared to the Foley catheter [14].

In our experience, employing a Foley catheter as a tourniquet to curtail hemorrhage during CS for PAS is a potential alternative. Yet, few publications currently explore this method. We found only four English-language articles on this subject, and they collectively discuss 23 patients who underwent perinatal hysterectomies due to PAS, using a Foley catheter as a tourniquet [15,16,17,18]. Our study aims to share our multi-year experience with the Foley catheter in reducing peripartum hemorrhage risks during hysterectomy for PAS patients.

## 2. Methods

### 2.1. Study Design and Settings

The study was conducted at the Department of Gynecology, Obstetrics, and Gynecological Oncology, Medical University of Silesia, Katowice, Poland. During the study period between 20 March 2017 and 30 March 2023, a total of 12 postpartum hysterectomies were performed due to PAS. In all cases, the Foley catheter was used as a tourniquet to reduce extensive bleeding. The method using the Foley catheter as a tourniquet in postpartum hemorrhage is a standard procedure in the department and has been approved by the Department’s Therapeutic Committee; first acceptance number: SOP/02/2016/GIN, date: 12 February 2016; and current acceptance number: SOP/01/2020/GIN, date: 20 January 2020. All twelve patients’ records were retrospectively studied and included in the present analysis. The collection and description of the data were prepared in accordance with the STROBE checklist.

### 2.2. Data Sources

Information was extracted on age, past medical history, gravidity, parity, number of previous CS, gestational age at delivery, indication for the current CS, obstetrical complications, fetal status, and previous assisted reproductive techniques. Additionally, data regarding intraoperative management were collected, which included the type of CS incision, type of anesthesia, uterine artery ligation, complications (such as bladder injury), and the quantity of packed red blood cells (PRBCs) and fresh frozen plasma (FFP) transfused. Blood absorbed by surgical gauze (Gauze Visual Analogue) and the volume of blood aspirated with a suction device were used to estimate perioperative blood loss. Postoperative data, such as postpartum complications, changes in hemoglobin levels, duration of hospitalization, and histopathological findings, were also included.

### 2.3. Participants

Every pregnant woman undergoing a cesarean section in our center due to PAS or suspected PAS is informed about the potential need for a peripartum hysterectomy. Patients are required to sign an informed consent for this procedure. The inclusion criteria for this analysis were a prenatal suspected diagnosis of placenta accreta spectrum and the performance of a peripartum hysterectomy. The exclusion criteria were patients who did not undergo a hysterectomy following the cesarean section or those who did not provide consent for the procedure.

### 2.4. Bias

Due to the rarity of the disease, which is PAS, and the need for peripartum hysterectomy, only 12 patients were included in the study. Another possible limitation is the lack of a control group to compare the outcomes.

### 2.5. Surgical Technique

The method used by the authors is an extension of the method proposed by Ikeda T et al. [19] and has been previously described in detail [20]. In brief, at the stage of planning the operation or in an emergency, the tourniquet method can be combined with other hemostatic methods, such as the embolization of the iliac vessels, hemostatic sutures, or the use of diathermy, but does not routinely assume utilization of any of the parallel procedures. We propose a modification of the Misgav-Ladach method [21]. Our method assumes the following stages:Skin incision: When conducting surgery for placenta accreta syndrome, a midline incision that avoids the umbilicus is preferred. If a primary lower segment incision is present, it should be extended upwards along the midline. The choice of skin incision technique should be based on the patient’s medical history and the surgeon’s experience and preferences.Access to the uterus: At this stage, we also use monopolar instruments to dissect the subcutaneous tissues. The rectus sheath is separated along its fibers. The rectus muscles are separated by pulling. The peritoneum is opened by stretching with index fingers.Opening the uterus: After gaining access to the abdominal cavity and visualizing the uterus, in cases of PAS, the uterine incision should be made above the intrauterine margins of the placenta to minimize bleeding. Prior to making the incision, it is advisable to perform an ultrasound to determine the optimal site for the uterine incision. The uterus is opened with an index finger and the hole enlarged between the index finger of one hand and the thumb on the other.Delivery of the babyEvaluation of the placenta and bleeding: After the baby is delivered, the uterus is extracted from the abdominal cavity, along with the placenta. In cases of a planned hysterectomy due to placenta accreta spectrum (PAS), the placenta is not detached from the uterus. If the placenta has been manually extracted and hemorrhage ensues, the procedure for inserting a Foley catheter remains the same, irrespective of whether the placenta remains in the uterus or not. The initial step involves releasing the uterine appendages. This is accomplished by manipulating the uterus in a horizontal manner. Subsequently, an assistant employs a sterile Foley catheter (Ch 16/18 French) to guide it caudally to the most inferior point and then secures it “en bloc” around the cervix (our technique avoids perforating the broad ligament) at the level of the uterosacral ligaments, approximately 3–4 cm below the incision line. Once positioned, the catheter is tightened and secured using Kocher forceps in preparation for the subsequent stages of the hysterectomy. The tourniquet technique facilitates hemostasis, granting the surgeon time to contemplate the potential for uterine preservation or the surgical approach to adhesions with neighboring organs. Given its straightforward and reversible placement, the tourniquet can be momentarily loosened to assess active bleeding, and then retightened to proceed with the operation. At this point, once the Foley catheter is clamped, both the surgical and anesthesia teams can ready themselves for subsequent phases of the procedure, especially if a hemorrhage occurs or if PAS is identified intraoperatively. Employing a Foley catheter as a tourniquet does not preclude the utilization of other techniques, encompassing both pharmacological and compression approaches. The specific surgical and Foley catheter insertion techniques are depicted in Figure 1 (graphical illustration) and Figure 2 (intraoperative image).Total vs. subtotal hysterectomy: In our center, the preferred method is the total removal of the uterus. Based on our experience and established scientific reports, total hysterectomy is associated with lower rates of reoperation and perioperative mortality and is less complicated than subtotal hysterectomy. We recommend retaining the cervix if hemorrhage can be effectively managed in this manner or if the surgeon is not confident in performing a total hysterectomy [22,23].Closing the Abdominal Wall: In our center, we consistently place abdominal drains following cesarean sections. For peripartum hysterectomies, we recommend inserting two drains—one above and one below the fascia. The rectus muscles are left unsutured. The fascia is closed using a continuous suture.Skin Closure: The skin can be closed using staples, sutures, or adhesive strips, depending on the surgeon’s preference.

### 2.6. Statistics

Data were entered into a spreadsheet. Statistical analyses were performed using computer software—the jamovi project (2022)—jamovi. (Version 2.3), retrieved from https://www.jamovi.org (accessed on 29 March 2022). The categorical variables are presented as both absolute numbers and percentages. After assessment of normality of distribution of continuous variables using Shapiro–Wilk test, the variables are presented as median with quartiles 1 and 3 for non-normal distribution and mean with standard deviation for normal distribution. Between-group comparisons of continuous variables were conducted using the Wilcoxon signed-rank test. The interval of two-sided *p* < 0.05 was considered statistically significant.

## 3. Results

### Participants, Descriptive Data and Periprocedural Characteristics and Outcomes

Twelve participants who underwent peripartum hysterectomy were included in the study. The median gestational age was 38 weeks (quartile 1–quartile 3: 36–38 weeks), and the mean age of the participants was 33.2 ± 3.9 years, with an age range of 29 to 40 years. Three patients (25.0%) were primiparous. None had significant comorbidities, although one had previously undergone cholecystectomy and another myomectomy. Seven patients (58.3%) had undergone a cesarean section (CS) in the past. One had undergone in vitro fertilization (IVF). Four received antenatal corticosteroids to mitigate risks of perinatal and neonatal death and respiratory distress syndrome. Patients 1 and 6 were administered antenatal corticosteroids before 34 weeks of gestation due to bleeding and risk of preterm delivery, while patients 7 and 9 received them in accordance with guidelines from the Polish Society of Gynecologists and Obstetricians and The Royal College of Obstetricians and Gynaecologists [24,25]. Despite undergoing a cesarean at 31 weeks of gestation, patient 11 did not receive antenatal corticosteroids because it was an emergency procedure. Every patient in the study underwent a cesarean section with subsequent postpartum hysterectomy due to PAS. Two required postpartum hysterectomy because of hemorrhage that necessitated relaparotomy (during the index hospitalization); conservative uterine preservation methods proved insufficient in these cases. In eight patients, the cesarean section was performed due to PAS, while two were diagnosed with PAS during surgery. The remaining two required surgery because of uterine atony and postpartum hemorrhage. All fetuses presented in the cephalic position, with a mean birth weight of 3050 ± 660 g. There were no fetal presentations other than cephalic. Four patients received antenatal betamethasone corticosteroid therapy. Table 1 summarizes the general characteristics of participants, indications for cesarean section, and obstetric outcomes.

No perioperative injuries were identified in any of the patients. There were no damages to the bladder, ureters, intestines, or other surrounding tissues. The average blood loss for all participants was 1200 ± 760 mL. Eight participants received PRBC units, and six received FFP units.

The median post-procedural length of hospital stay was 5 days (quartile 1–quartile 3: 4–6 days). The longest stay was 16 days and the shortest was 2 days. The two extended stays (*) resulted from mothers awaiting their newborns, due to premature delivery. Nonetheless, if we consider discharging these patients earlier without their child, the average hospital stay would be approximately 2.4 days. One patient had a cesarean section in the 31st week of pregnancy, and the newborn was transferred to the Neonatal Intensive Care Unit in another hospital.

Table 2 summarizes the intraoperative management, complications, length of stay, preoperative and postoperative hemoglobin levels, postpartum complications, and histopathological diagnoses for each patient.

The mean preoperative Hb was 12.8 ± 1.6 g/dL (range: 10.0 g/dL to 14.4 g/dL). The mean postoperative Hb was 8.7 ± 2.3 g/dL (range: 4.9 g/dL to 12.0 g/dL), *p* < 0.001. The average hemoglobin loss was 3.5 g/dL. The histopathological examination confirmed PAS in all cases. The average duration of the procedure was 89 ± 17 min. A summary of this study can be found in Figure 3.

## 4. Discussion

The management of severe postpartum hemorrhage (PPH) necessitating peripartum hysterectomy presents a formidable challenge in obstetrics, especially when associated with the placenta accreta spectrum. This is among the numerous difficulties that can arise during a cesarean section, and stands out as one of the most severe complications [26,27].

The use of a Foley catheter as a tourniquet during peripartum hysterectomy has emerged as a promising intervention for controlling hemorrhage in these intricate cases. Historically, a variety of strategies were introduced to handle excessive bleeding during peripartum hemorrhages, encompassing both nonsurgical techniques and surgical interventions such as uterine arterial ligation, embolization, and hemostatic suturing [8,9,10,18]. It is important to highlight that each method comes with its own set of limitations and risks. Consequently, in recent years, we have adopted the Foley catheter as our standard approach to mitigate bleeding in patients with PAS undergoing peripartum hysterectomy. Our research primarily delves into the placenta accreta spectrum (PAS) and perinatal hysterectomy.

To date, no other publication on this topic matches the scale of our study, making it the largest English-language publication on the subject. We identified only four English-language publications, and together, they encompassed 23 patients who underwent perinatal hysterectomies due to PAS where a Foley catheter was used as a tourniquet [9,10,11,12]. The recurring exploration of tourniquets in these studies emphasizes the perceived potential of this technique in controlling hemorrhage during cesarean sections. While methodologies differ, the aggregated data suggest that tourniquets, either alone or combined with other techniques, may present a viable solution to this clinical dilemma. Still, questions about the safety, efficacy, and potential side effects of extended tourniquet use persist, highlighting the need for additional trials and evaluations.

Abdelaziz et al. [15] presented a unique tourniquet specifically designed to reduce blood loss during surgical treatment of PPH in cesarean deliveries. While specific details on the design or function of this tourniquet are not elaborated upon, it hints at potentially promising outcomes. Meng et al. [16] proposed using two tourniquets in succession, suggesting a systematic strategy to manage bleeding from multiple sources and possibly minimizing risks tied to prolonged use of a single tourniquet. Huang et al. [17] explored the combined use of tourniquets and forceps for vascular control in cases of placenta accreta spectrum. Merging mechanical (tourniquet) and manual (forceps) techniques suggests a comprehensive approach to bleeding management, though it raises questions about potential tissue trauma or extended surgery duration. Altal et al. [18] conducted a pilot study assessing the efficacy of a cervical tourniquet during cesarean deliveries in cases with morbidly adherent placenta. By centering on the cervix, this study illuminates a novel approach that could specifically address a primary source of bleeding. However, given its pilot status, the study might encompass a limited sample, underscoring the need for more extensive research to ascertain the method’s safety and effectiveness.

Our study adds to the body of knowledge by further substantiating evidence that supports the feasibility, low risk of perioperative bleeding, and the periprocedural safety of this approach. Our results show that the systematic use of a Foley catheter in hysterectomy cases due to PAS can yield satisfactory clinical outcomes. This is corroborated by the average periprocedural blood loss, which was measured at 1200 ± 760 mL. The mean number of transfused packed red blood cell (PRBC) units stood at 2 ± 2. Given the standard volume of an RBC unit, only two patients required a transfusion of more than one liter of PRBC.

The total number of cases in our study, amounting to 12 hysterectomies due to PAS, should be noted. While some might view this as a limitation, it also emphasizes the relatively low occurrence of this serious clinical condition. It is worth noting that almost 12,000 newborns were delivered in our department during the period of this study.

While a direct comparison with other methods is not feasible, previous studies employing compression techniques reported equal or greater blood loss [10].

One undeniable advantage of the single Foley catheter tourniquet method is its widespread availability and cost-effectiveness. This tool is fundamental in surgical departments and ranks among the most affordable equipment. Furthermore, the method does not necessitate additional training, boasts high reliability and repeatability, and its typical construction—often a latex core covered in either hydrogel or a silicone-elastomer coating—ensures adequate pressure to effectively compress the bleeding uterus. Prompt compression is crucial, as it offers the surgeon a clear view of the operative area, thereby enhancing surgical comfort and potentially reducing the procedure duration.

In our cases, the average procedural duration was approximately 90 min. This suggests a relatively short surgical time, considering the complexity of the procedure. Another point to note is the postoperative stay duration, with a median of 5 days. This duration includes two patients who remained in the hospital to await their newborns, whose stays were extended due to prematurity. If we exclude these two patients, the average postoperative stay would have been 4 days. Importantly, none of our patients required subsequent surgical intervention post hysterectomy, underscoring the effectiveness of our approach to manage perioperative bleeding.

Our study, while presenting the results of the largest cohort of peripartum hysterectomies managed with a Foley catheter tourniquet to date, has several notable limitations. Firstly, the number of cases is still relatively small, and the study design is retrospective. Therefore, direct conclusions and assertions of causality should be made with caution. Secondly, while our study indicates successful outcomes in all patients, there might be specific clinical situations where the Foley catheter technique may not be as effective. These scenarios could include, for example, anatomical uterine variations, complexities in the local operative field, or cases of low-lying placenta previa, which could complicate securing the catheter around the uterus’s lowest point. To our knowledge, however, no such instances have been documented thus far. Lastly, patients with a latex allergy might potentially react to the Foley catheter. In such instances, an alternative approach may be warranted [28].

## 5. Conclusions

In conclusion, using a Foley catheter as a tourniquet on the lower segment of the uterus may emerge as the method of choice to prevent peripartum hemorrhage in patients with placenta accreta spectrum (PAS). The catheter’s adjustable pressure provides precise occlusion of blood vessels, promoting rapid hemorrhage control and potentially reducing the need for more invasive surgical procedures.

The widespread availability and cost-effectiveness of the Foley catheter make it a valuable choice in diverse healthcare environments. By reducing intraoperative blood loss, this method not only improves the surgical field but also enhances postoperative outcomes, paving the way for a faster recovery and earlier hospital discharge. This technique is not only straightforward but also cost-effective. It is important to note that should hemorrhage or complications related to PAS arise, this method is instrumental in minimizing blood loss, giving the surgical and anesthesia teams essential time to plan further life-saving interventions. By introducing an innovative approach to tackle the challenges of hemorrhage during peripartum hysterectomy associated with PAS, our study aims to enhance patient outcomes, reduce maternal morbidity and mortality, and elevate surgical standards.

## Figures and Tables

**Figure 1 life-13-01774-f001:**
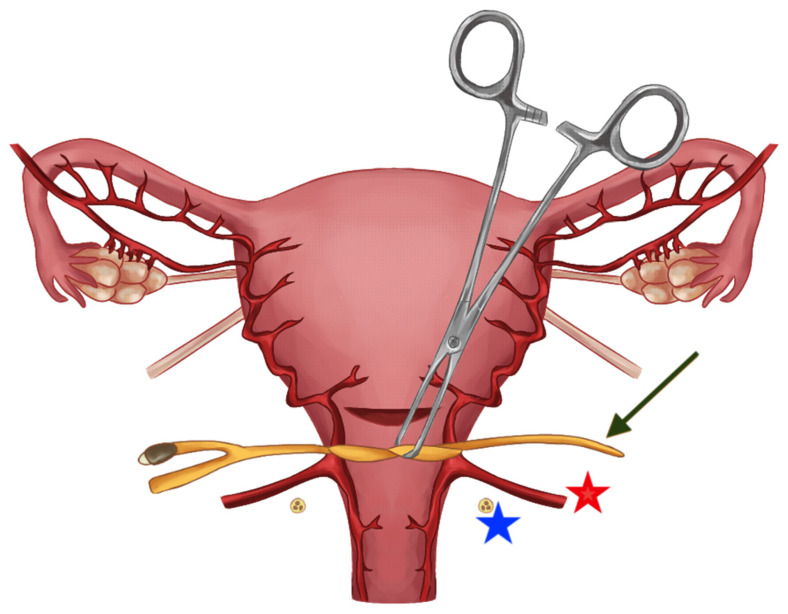
Graphical illustration of the method. The Foley catheter (indicated with a green arrow) is knotted below the incision line on the uterus, and then tightened with Kocher forceps in order to reduce the periprocedural bleeding from the uterine vessels (indicated with a red star). The ureter (indicated with a blue star) can in most cases be avoided by the method.

**Figure 2 life-13-01774-f002:**
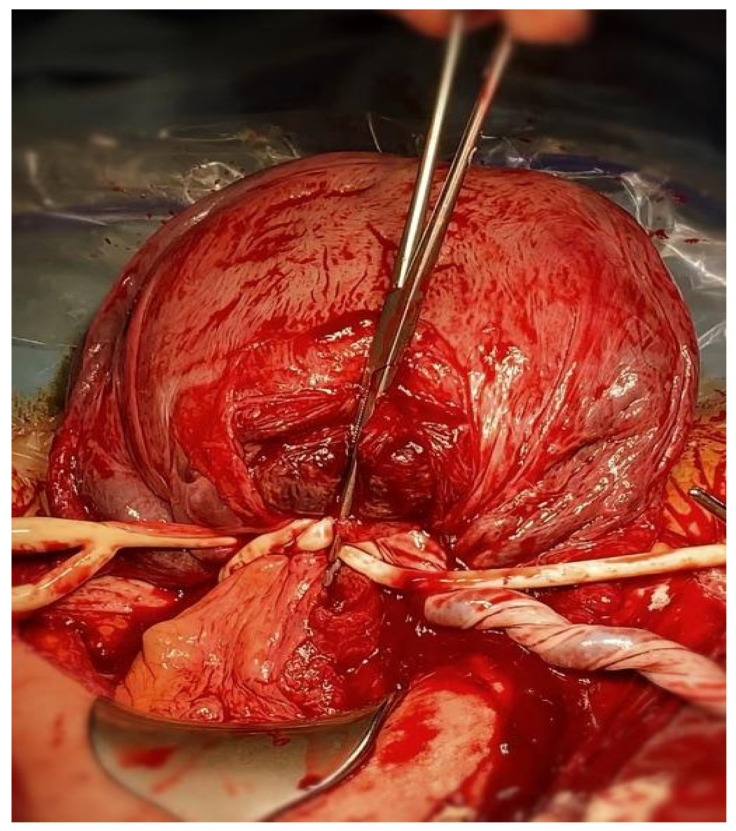
Intra-surgery image of Foley catheter localization.

**Figure 3 life-13-01774-f003:**
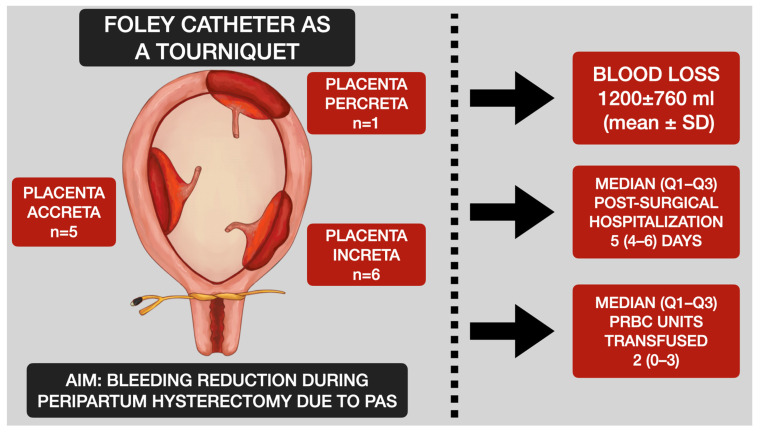
Central illustration: The procedural scheme and outcomes in patients undergoing Foley catheter as a tourniquet for hemorrhage prevention during peripartum hysterectomy in patients with placenta accreta spectrum.

**Table 1 life-13-01774-t001:** General characteristics of participants, indications for cesarean section, and obstetric results.

Patient	Maternal Age (Years)	Gravidity	Parity	Previous CS (n)	Gestational Age (Weeks)	Associated Conditions	Cause of CS	Birth Weight (g)	APGAR Score
1	30	3	3	2	37	None	Placenta percreta, 2 previous CS	3210	9
2	38	3	3	1	38	None	Placenta increta, 1 previous CS	3070	8
3	32	1	1	0	38	PIH	Fetal tachycardia/relaparotomy (Uterine atony)	3650	10
4	37	1	1	0	38	Lumbosacral discopathy, IVF	Lumbosacral discopathy/relaparotomy (Uterine atony)	3350	10
5	28	2	2	0	39	None	Marginal placenta praevia	3830	10
6	30	1	1	0	38	None	Marginal placenta praevia	3270	10
7	36	2	2	1	34	Myomectomy	Placenta increta	2470	9
8	31	2	1	0	38	None	Placenta praevia	3430	10
9	32	3	2	1	33	None	Placenta praevia, 1 previous CS	2130	9
10	35	3	3	2	37	Hypothyroidism	PROM, 2 previous CS	2880	10
11	29	3	2	1	31	Cholecystectomy	Placenta praevia, antepartum hemorrhage	1660	8
12	40	2	2	1	40	None	Ophthalmic, 2 previous CS	3660	10

**Table 2 life-13-01774-t002:** Summary of the intraoperative management and complications.

Patient	CS Insicion	Anesthesia	Hysterectomy	Operation Time (min)	Preoperative Hb (g/dL)	Postoperative Hb (g/dL)	Blood Loss (mL)	PRBC (n)	FFP (n)	Length of Stay (Days)	Histopathological Findings
1	Midline vertical incision	SA	Total	109	10.0	10.8 *	400	2	0	5	Placenta percreta
2	Midline vertical incision	SA	Total	86	12.8	8.4	1200	0	0	5	Placenta increta
3	Low transverse cesarean section	SA/GA (relaparotomy)	Total	111	10,6	5.4	2000	5	2	4	Placenta accreta
4	Low transverse cesarean section	SA/GA (relaparotomy)	Total	71	12.5	5.5	2100	6	2	6	Placenta increta
5	Low transverse cesarean section	SA	Total	86	14.3	12.0	800	2	0	4	Placenta accreta
6	Low transverse cesarean section	SA	Total	113	13.5	9.6	1300	2	2	4	Placenta accreta
7	Low transverse cesarean section	SA	Total	80	12.6	9.7	650	0	0	l0 *	Placenta increta
8	Low transverse cesarean section	GA	Total	89	10.8	4.9	2500	3	2	5	Placenta increta
9	Midline vertical incision	GA	Total	110	12.0	9.3	1200	2	1	16 *	Placenta accreta
10	Low transverse cesarean section	SA	Total	77	15.2	10.9	2600	0	0	2	Placenta increta
11	Midline vertical incision	GA	Total	78	12.3	8.2	900	3	2	3	Placenta increta
12	Low transverse cesarean section	SA	Subtotal	58	10.7	9.9	600	0	0	3	Placenta accreta

* indicates the first laboratory value of hemoglobin after peri-operative transfusion of 2 PRBC units.

## Data Availability

Further data on patients included in this study can be made available on request to the corresponding author.

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
