# Peer review of "Foley Catheter as a Tourniquet for Hemorrhage Prevention during Peripartum Hysterectomy in Patients with Placenta Accreta Spectrum (PAS)—A Hospital-Based Study"

_life, 2023, doi:10.3390/life13081774_

Round 1

Reviewer 1 Report

Dear authors,

thank you for sending „Foley catheter as a tourniquet for hemorrhage prevention during peripartum hysterectomy in patients with placenta accreta spectrum (PAS) - A hospital-based study“ to Life.

This is an interesting single-center study presenting data about the use of a Foley catheter for prevention of peripartal hemorrhage in the relatively rare event of PAS. The study group is quite big. PPH-therapy is still an unsolved problem worldwide, every new idea of improving PPH-management should be evaluated and compared to existing standards of care.

I would recommend publication with major revision having some questions and comments:

I wonder why the authors limit the use of a foley catheter to patients with PAS. Do you only use this technique in those patients or also in other circumstances to stop bleeding?

I do not understand why you decided for hysterectomy in all of these 12 cases? Theoretically PAS is not an obligatory indication for hysterectomy, even in patients with placenta percreta the uterus may be preserved if the patient stays clinically stable which was the case in all of your patients apparently. Please comment on that.

Introduction:
First paragraph: please provide literature for your claims.
Line 30: The main reason for PPH is atony, vessels ligation is not the first and most important therapy for atonic bleeding. If uterotonic drugs and tamponades do not work, vessel ligation is used. In PAS-patients the situation is different and vessels ligation may be the first step of treatment - please specify this difference here.
Line 44: please provide literature
Line 52, 54: source of your claims?
Line 53: there is also the possibility to clamp the uterine vessels temporarily or do an uterine artery ligation. This is however a quite difficult and risky method. Please add these options to the mentioned possibilities or discuss it respectively.
Line 60: literature?
Line 67: which studies?
Line 69: several? Please specify

Methods:
Line 88: also gestational weeks at delivery
Line 91: how was your preoperative management? Based on which information did you decide the type of CS incision? Do you regularly do MRI for planning?
Line 101: Do you discuss other therapy options with the patients such as partly resection of the affected part of the uterus or leaving of the placenta in situ instead of hysterectomy?
Line 102: you mean suspicion of PAS? Prenatal diagnosis of eg placenta accreta is very rarely possible from my point of view. Your diagnosis is based on which imaging?
Line 142: What do you mean by „between-group comparisons of continuous variables“? I did not find these results? Which groups do you mean?

Results:
Paragraph 3.1: is almost completely a repetition of table 1. Please decide for one kind of presentation of your data, either text or table
Line 155: In the two patients requiring postpartum relaparotomy due to atonic hemorrhage PAS was not the reason for hysterectomy, wasn’t it? Please provide more data about these two cases, which patient number, how did you remove the placenta etc?
Line 167: why did those 4 patients receive bethamethasone, please specify. There is no need to give double information: if all fetuses were in cephalic position you do not have to have the parameter „fetus presentation“ in your table.
Table 2: please bring together table 1 and 2 for better understanding. Is the column „obstetric complications“ really necessary?, this information becomes clear from context.
Table 3: please bring together table 3 and 4 for better understanding. The column „perioperative injury“ is not necessary as you provide information about that in the text.
Based on which informations you decide to do either a transverse or a vertical incision?
Why did so many patients receive so many PRBCs if they had no severe blood loss? Do you give PRBCs prophylactically? PPH is defined as >1000ml blood loss in C-sections, according to guidelines.
Line 174: 1200ml - 760ml blood loss is not a PPH and no indication for hysterectomy!?
Line 178-181: is this a table caption or normal text?
Line 179: what do you mean by this p-value? What is significant? The blood loss? There is no need for this specification.
Line 180: please try to avoid double information (tables/text)
Table 4: column „postpartum complications“ seems to be redundant
Figure 2: gives no new information and can be removed accordingly

Discussion:
Line 208-222: the second paragraph of the discussion is just a general explanation about PAS and may be skipped or transferred to introduction section. In the discussion the subject and the results of the paper should be discussed.
Line 238: please discuss the published paper on your subject in more detail instead of presenting table 5 only. Table 5 may be removed then.
Line 240: safety and effect of this quite new method has been shown several times. Why should your paper be published additionally, which new findings do you provide? Please specify. A control group would have been important, indeed. Perhaps a project for future research?
Line 246: I cannot estimate if 12 hysterectomies are really a low frequent event in your institution. For that I would need the whole number of deliveries in the study’s time period. Please provide.

Generally, literature references are insufficient, especially in the introduction.

Thank you.

Best regards

There are plenty of language mistakes. Please present the manuscript to a native-speaker before application.

Author Response

Dear Reviewer,

Thank you for your comprehensive review and feedback on our manuscript titled "Foley catheter as a tourniquet for hemorrhage prevention during peripartum hysterectomy in patients with placenta accreta spectrum (PAS) - A hospital-based study" submitted to Life. We genuinely appreciate the time and effort you have invested in providing such detailed feedback, which will undoubtedly enhance the quality of our work.

Please find our responses to your questions and comments:

I wonder why the authors limit the use of a foley catheter to patients with PAS. Do you only use this technique in those patients or also in other circumstances to stop bleeding?

We are grateful for this remark, and the entire review. According to our Department’s standards, we use the Foley catheter as a tourniquet to prevent postpartum hemorrhage, regardless of the cause of the hemorrhage, not only in patients with PAS. However, the aim of this manuscript has been to summarize the outcomes of a relatively homogenous group of patients - thus, patients with PAS were selected, due to the significance of this condition, and the difficulties in manual management of bleeding (due to deep penetration by the placental tissue of both the uterus, and the adjacent organs). Therefore, the subgroup of patients with PAS was selected.

I do not understand why you decided for hysterectomy in all of these 12 cases? Theoretically PAS is not an obligatory indication for hysterectomy, even in patients with placenta percreta the uterus may be preserved if the patient stays clinically stable which was the case in all of your patients apparently. Please comment on that.

We would like to thank for this remark. Indeed, PAS is not considered as an obligation to perform hysterectomy. Therefore, in our Department, in patients diagnosed with PAS we have implemented a strategy to consider conservative treatment and preservation of the uterus as the treatment of choice in all eligible patients. The purpose of this particular study was not to evaluate the overall cohort of patients with PAS, but to focus specifically on those, in whom hysterectomy was necessary due to medical conditions. The described group included patients in whom excessive blood loss resulted in hemodynamic instability, which could not be controlled and thus, hysterectomy was obligatory. Among the patients mentioned  were also ones in whom, thanks to the use of a Foley catheter as a tourniquet , the bleeding was not so profuse. In those cases, after analyzing the ultrasound and MR images performed during pregnancy indicating a significant degree of placental ingrowth, as well as the patient’s prior preferences regarding the future maternity plans, a decision on the hysterectomy was made by the team.

Line 30: The main reason for PPH is atony, vessels ligation is not the first and most important therapy for atonic bleeding. If uterotonic drugs and tamponades do not work, vessel ligation is used. In PAS-patients the situation is different and vessels ligation may be the first step of treatment - please specify this difference here.

We do agree with the thesis stated by the Reviewer. The mechanism leading to PPH in patients with uterus atony is substantially different from the one in PAS. For instance, in patients with atony, the primary cause of bleeding is insufficient contraction of the uterus, which could be managed by the use of contractility-supporting strategies, e.g. uterotonic drugs. In PAS, the bleeding is a result of pathologically deep invasion of placental tissue into the uterus and the adjacent organs. In such situation, the uteral contractility is not mitigated to the extent as in atony, thus the uterotonic drugs and tamponades might not bring sufficiently satisfying results. 

Nonetheless, the purpose of the manuscript has not been to evaluate the entire population of patients with PPH, but to focus more specifically on patients with PAS. In such patients, as already elaborated in the response above, our primary strategy has been to consider conservative treatment and preservation of the uterus as the treatment of choice (always when applicable), although the manuscript presentes thoroughly described 12 cases, in whom hysterectomy was mandatory due to excessive bleeding. 

Line 91: how was your preoperative management? Based on which information did you decide the type of CS incision? Do you regularly do MRI for planning?

We would like to thank for this important remark. We do agree with this important subject, which has already at least partially been discussed in a response to Reviewer #2. The location of the incision largely affects the further course of the surgery, and potentially influence both operators’ comfort, but also the risk of periprocedural complications. However, in this regard, the location of the incision does not reflect the incision of the uterus, but of the skin. This information reflects both the location of the placenta and operator preference. Nonetheless, as we do agree that such presentation could mislead the readers, we have added the word “abdominal” in the column title, which now stands as “CS abdominal incision”

With regard to the procedural planning with MRI, our protocol usually involves the verification of an ultrasound examination with an MRI examination. However, this test is recommended, not mandatory, especially in patients presenting in an acute setting, who had PAS diagnosed outside of our facility. In such patients, the necessity to implement a dynamic, appropriate response to the clinical condition often did not allow to perform MRI. 

Line 101: Do you discuss other therapy options with the patients such as partly resection of the affected part of the uterus or leaving of the placenta in situ instead of hysterectomy?

We would like to thank for this remark. The prior discussion with the patient is essential to provide patient-centered care, and engage patient in obtaining informed consent, via patient’s active participation in the decision-making process. Whenever it is possible (provided not in an emergency condition) each patient is to be provided with an elaborate information about every possible surgical technique, including both fertility-saving and radical approaches. It is of utmost importance that each PAS surgery is tailored to the patient's expectations, her health condition and the anticipated course of the surgery - namely, after thorough analyses of both ultrasound scans, as well as the MRI images (whenever possible), taking into consideration the depth of tissue invasion, the penetration of adjacent organs, the overall clinical condition of every particular patient, a forecast is made on the possibility of uterus-preserving techniques. However, each patient is informed about the risks associated with any technique, and is aware of the possibility of the necessity to crossover from the initially assumed strategy.

Line 102: you mean suspicion of PAS? Prenatal diagnosis of eg placenta accreta is very rarely possible from my point of view. Your diagnosis is based on which imaging?

We are grateful for this question and comment. We completely agree with the Reviewer that the definite diagnosis of PAS is often difficult, and the final diagnosis can be made based on the histopathological examination post-partum. Therefore, the initial statement could have potentially introduced some degree of confusion for the future readers. Thus, we have modified the phrase, which now states “Every pregnant woman undergoing a cesarean section in our Center due to PAS or suspicion of PAS is informed about the potential need for a peripartum hysterectomy.” 

Line 142: What do you mean by „between-group comparisons of continuous variables“? I did not find these results? Which groups do you mean?

We are thankful for this comment. We agree with the Reviewer that throughout the text no comparisons between independent variables exist, and thus U-Mann Whitney test could not have been used. In this case, it has been a writing mistake, since the test used for comparison of continuous variables was Wilcoxon signed-rank test, which allowed us to compare Hemoglobin levels prior to and after surgery. The change in the text has been implemented accordingly,

Results:

Paragraph 3.1: is almost completely a repetition of table 1. Please decide for one kind of presentation of your data, either text or table

We would like to thank the Reviewer for this important remark. The purpose of detailed presentation of data in Paragraph 3.1 was to emphasize the most important information on the clinical characteristics of the studied group. The information presented in the paragraph concern age, medical history, and most critical aspects of peripartum strategy, while the details for every patient are scrupulously expounded in Table 1. Therefore, we do believe that despite significant word reduction, and the mitigation of redundant wording, this particle paragraph should remain. 

Line 155: In the two patients requiring postpartum relaparotomy due to atonic hemorrhage PAS was not the reason for hysterectomy, wasn’t it? Please provide more data about these two cases, which patient number, how did you remove the placenta etc?

We would like to thank the Reviewer for this important remark. We do believe that the situation mentioned by the Reviewer greatly fits into the strategy utilized in our centre. As stated above,  It is of utmost importance that each PAS surgery is tailored to the patient's expectations, health condition and the anticipated course of the surgery - in those two cases (number 3 and number 4), the uterus-preserving technique has been initially implemented after discussion with the patient, and therefore, during C-section, we tried to preserve the uterus with focal excision of PAS but because of postpartum hemorrhage and uterine atony after the surgery, the relaparotomy and eventually hysterectomy was necessary. The diagnosis of PAS was confirmed after histopathological examination. Those cases demonstrate the necessity of adequate qualification of the patients, and discussion the expectations and limitations of each technique with each patient.  

Line 167: why did those 4 patients receive bethamethasone, please specify. There is no need to give double information: if all fetuses were in cephalic position you do not have to have the parameter „fetus presentation“ in your table.

We would like to thank for this question. Four patients received antenatal corticosteroids to reduce perinatal and neonatal death and respiratory distress syndrome. Patients 1 and 6 received antenatal corticosteroids before 34 weeks of gestation due to bleeding and the risk of preterm delivery, during previous hospitalizations. Patients 7 and 9, whose caesarean sections were performed in the 34th and 33rd week of pregnancy, received antenatal corticosteroids in accordance with the recommendations of the Polish Society of Gynecologists and Obstetricians and The Royal College of Obstetricians and Gynaecologists. Patient 11, despite the fact that the caesarean section was performed in the 31st week of pregnancy, did not receive antenatal corticosteroids due to emergency caesarean section. In our clinic we have introduced preference for betamethasone and we are satisfied by the outcomes of this steroid drug. Finally, we agree with the redundancy of cephalic position presentation in the text, therefore we have removed the column “fetus presentation” from the Table 2. 

Table 2: please bring together table 1 and 2 for better understanding. Is the column „obstetric complications“ really necessary?, this information becomes clear from context.

We would like to thank the Reviewer for this remark. We have removed the column “obstetric complications” and combined information from Table 1 and Table 2 - we believe that after such change, the manuscript is more concise, and more information is succinctly presented in a combined table. 

Table 3: please bring together table 3 and 4 for better understanding. The column „perioperative injury“ is not necessary as you provide information about that in the text.

We are thankful for this remark.  We have removed the column “perioperative injury” and combined information from Table 3 and Table 4. As already stated above, we believe that after such change, the manuscript is more concise, and more information is succinctly presented in a combined table.

Based on which informations you decide to do either a transverse or a vertical incision?

We would like to thank for this important remark. However, as the subject has already been brought to light by the Reviewer, as well as Reviewer #2, we will allow ourselves to present the exact response, to avoid redundant wording.

The location of the incision largely affects the further course of the surgery, and potentially influence both operators’ comfort, but also the risk of periprocedural complications. However, in this regard, the location of the incision does not reflect the incision of the uterus, but of the skin. This information reflects both the location of the placenta and operator preference. Nonetheless, as we do agree that such presentation could mislead the readers, we have added the word “abdominal” in the column title, which now stands as “CS abdominal incision”

With regard to the procedural planning with MRI, our protocol usually involves the verification of an ultrasound examination with an MRI examination. However, this test is recommended, not mandatory, especially in patients presenting in an acute setting, who had PAS diagnosed outside of our facility. In such patients, the necessity to implement a dynamic, appropriate response to the clinical condition often did not allow to perform MRI. 

Why did so many patients receive so many PRBCs if they had no severe blood loss? Do you give PRBCs prophylactically? PPH is defined as >1000ml blood loss in C-sections, according to guidelines.

We would like to thank the Reviewer for this important question and comment. The patients with PAS, or suspected PAS are at elevated risk of bleeding, not negligibly even critical. Thus, before each cesarean section of a patient with suspected PAS, at least 2 blood units were reserved and cross-matched, and this number could be adjusted based on patient’s condition, and the risk of bleeding at the operator’s discretion. We do agree that in some of cases presented in the table, the overall blood loss is lower than 1000 ml, which would normally not be considered as clinically meaningful. However, few important information need to be taken into account. First, the decision to transfuse the PRBC is not solely based on the absolute loss, but also on the absolute hemoglobin levels, which often are low even before the procedure. Second, a dynamics of loss is important, since in some patients the drop of blood loss might be less rapid, and in some much more dynamic. Finally, the decisions in the OR are always team-based, thus the recommendations of the anesthesia team need to be taken into account. 

A recurrent analysis of all cases has indicated that in one patient (no 1), the value labelled as
“Postoperative Hb (g/dl)” was the first result obtained after peri-operative transfusion of 2 PRBC units”., therefore it could be speculated that in that patient, the postoperative Hb level without transfusion would have been much lower. Thus, the following information has been presented below the table “Caption: * indicates the first laboratory value of hemoglobin after peri-operative transfusion of 2 PRBC units.” 

Line 174: 1200ml - 760ml blood loss is not a PPH and no indication for hysterectomy!?

We would like to thank the Reviewer for this remark. We do agree that the absolute blood loss should not be, and in that case, was not, the sole cause for deciding on hysterectomy. As already stated in the prior responses, among the patients included in the analysis, the were also those in whom, thanks to the use of a Foley catheter as a tourniquet , the bleeding was not so profuse. In those cases, after analyzing the ultrasound and MR images performed during pregnancy indicating a significant degree of placental ingrowth, as well as the patient’s prior preferences regarding the future maternity plans, a decision on the hysterectomy was made by the team. It could be once more speculated that without the use of a Foley catheter as a tourniquet, the blood loss would have been much greater. 

Line 178-181: is this a table caption or normal text?

We apologize for any uncertainty or misunderstanding which resulted from editing process prior to submission. The lines included a normal text, and appropriate changes were provided  in the manuscript. 

Line 179: what do you mean by this p-value? What is significant? The blood loss? There is no need for this specification.

We are grateful for this remark. The p-value refers to the already discussed comparison of continuous variables with the Wilcoxon signed-rank test, which allowed us to compare Hemoglobin levels prior to and after surgery. Nonetheless, we do agree with the Reviewer that the p value refers to the between-group paired comparison, not to the absolute value per se. Thus, the change in the text has been implemented accordingly and now the modified text states “The mean preoperative Hb was 12.8 ± 1.6 g/dl (range: 10.0 g/dl to 14.4 g/dl). The mean postoperative Hb was 8.7 ± 2.3 g/dl (range: 4.9 g/dl to 12.0 g/dl), p<0.001. “

Line 180: please try to avoid double information (tables/text). Table 4: column „postpartum complications“ seems to be redundant . Figure 2: gives no new information and can be removed accordingly

Thank you for the comments regarding the need to modify the structure of our manuscript. We do agree that for the clarity of the text it is mandatory to avoid redundant wording. Therefore, the changes were made, including the deletion of repetitive text regarding information presented in the Tables, deletion of the recommended column. We believe that after such change, the manuscript is more concise, and more information is succinctly presented in two combined tables. 

However, with regard to the Figure, we would like to kindly disagree, since in our opinion the figure has both educational, and illustrative merits, which would be hardly achievable solely by the text. We have strong confidence that an illustrative “graphical abstract” allows to grasp all necessary information at a glance, attracting attention, while maintaining scientific quality and detail. Thus, we would like the figure to be spared in the text.

Line 246: I cannot estimate if 12 hysterectomies are really a low frequent event in your institution. For that I would need the whole number of deliveries in the study’s time period. Please provide.

We would like to thank the Reviewer for this remark. We do strongly agree that at present, no exact information on the occurrence of PAS requiring hysterectomy is available. Therefore, we have analysed the exact number of deliveries in the studied period (20 March 2017 – 30 March 2023). The information confirms our baseline assumptions that prevalence of PAS requiring hysterectomy is low. Thus, we have added the following phrase to the manuscript, which states: “The total number of cases in our study, amounting to 12 hysterectomies due to PAS, should be noted. While some might view this as a limitation, it also emphasizes the relatively low occurrence of this serious clinical condition. It's worth noting that almost 12,000 newborns were delivered in our Department during the period of this study.” We hope that such change will allow to evaluate the problem by the future readers more appropriately. 

We are committed to addressing all the concerns you've raised and making the necessary revisions to improve our manuscript. We hope that our revised manuscript will meet the standards of Life and contribute meaningfully to the scientific community.

Once again, thank you for your invaluable feedback.

Best regards,

Jakub Staniczek

Reviewer 2 Report

Thank you for allowing me to read the manuscript on a very important topic.

I have some observation to make:

- the study group is very small even for such a are pathology and the main flaw is that it's not compared to other methods in order to prove the efficacy

Describing the method placing the Foley is unclear an one should  not rely upon the fact that the reader is familiar to the technique proposed by Ikeda. 

An intrasurgery image would be more informative.

in Table 1 "gravity" must be replaced with gravidity or synonime  and "medical illness" with associated  condition

in Table 2 there are some patients that required bethametasone but according to the baby's weight there is no reason, could you explain?

In table 3 assuming that the patients are identical to table 2 there is a question about why performing low segmental CS in cases of placenta praevia or scarr. Transsection of the placenta could complicate the surgery an increase the blood loss.

I suggest in order to support your conclusion to compare the method parameters (blood loss, transfusion etc) with a hystorical similar lot.

english must be improved, there are some words that are misused

Author Response

Dear Reviewer, 

Thank you for taking the time to review our manuscript and for providing valuable feedback. We appreciate your insights and constructive comments, which will undoubtedly help improve the quality and clarity of our work.

Regarding your observations:

The study group is very small even for such a are pathology and the main flaw is that it's not compared to other methods in order to prove the efficacy

We would like to thank you for this remark. Indeed, we do believe that the subject is of extreme importance, as both our clinical practice, as well as global data on the increasing rate of risk factors for PAS, indicate that in the future, the number of patients affected with this condition should be increasing. Therefore, identification of methods which allow to prevent critical bleeding is of extreme importance. 

We do agree with the Reviewer on the relatively low absolute number of cases included in the present manuscript, although we would like to emphasize few important conditions, which need to be taken into consideration while analyzing its results:

1) Although the real, relative frequency of PAS is hardly accessible, the prior reports indicate that its occurrence might be approximately up to 1 in 50 if there has been one cesarean section in the past - thus, it could be estimated that the occurrence of PAS in overall cohort of pregnant women could be approx. 1 in 100. Among them, the risk of severe bleeding is lower than 5%, therefore, among 10000 pregnant women, approximately 5 might be at risk of hemorrhage due to PAS. 

2) Our results demonstrate the efficacy and safety of a systematic approach to patients with bleeding due to PAS - therefore, although the absolute number of cases is 12, in all cases, a completely similar procedure was performed, which mitigates the risk of irreproduciblity of data in larger populations.

Moreover, our assertions are partially supported by the Reviewer #1, who has emphasized the large number of cases included in the present analysis. Therefore, we do consider our studied group as sufficiently numbered to present preliminary data on their outcome. 

3) Our results should also be considered, as of the first, hypothesis generating analysis, which could potentially allow for a wider adoption of the technique

Nonetheless, as we do consider the comment of the Reviewer as of importance, we have added the following sentence to the text: 

"Nevertheless, it must be emphasized that the overall occurrence of severe bleeding due to PAS in pregnancy is relatively low, thus the 12 patients represent our multiple years of experience and practice."

Describing the method placing the Foley is unclear an one should  not rely upon the fact that the reader is familiar to the technique proposed by Ikeda. 

An intrasurgery image would be more informative.

We are thankful for this Remark, as well as for the entire Review. We have added the intraoperative image to the manuscript. We hope that the added image, as well as the previously created and hand-drawn scheme would allow to more appropriately catch the attention of the future readers and increase the knowledge on the field and method.

In Table 1 "gravity" must be replaced with gravidity or synonime  and "medical illness" with associated  condition

We would like to thank you for this comment, and apologize for the prior spelling mistakes. We have modified the table accordingly. 

In Table 2 there are some patients that required bethametasone but according to the baby's weight there is no reason, could you explain?

We would like to thank for this question. Four patients received antenatal corticosteroids to reduce perinatal and neonatal death and respiratory distress syndrome. Patients 1 and 6 received antenatal corticosteroids before 34 weeks of gestation due to bleeding and the risk of preterm delivery, during previous hospitalizations. Patients 7 and 9, whose caesarean sections were performed in the 34th and 33rd week of pregnancy, received antenatal corticosteroids in accordance with the recommendations of the Polish Society of Gynecologists and Obstetricians and The Royal College of Obstetricians and Gynaecologists. Patient 11, despite the fact that the caesarean section was performed in the 31st week of pregnancy, did not receive antenatal corticosteroids due to emergency caesarean section. In our clinic we have introduced preference for betamethasone and we are satisfied by the outcomes of this steroid drug. 

In table 3 assuming that the patients are identical to table 2 there is a question about why performing low segmental CS in cases of placenta praevia or scarr. Transsection of the placenta could complicate the surgery an increase the blood loss.

We would like to thank for this comment. Indeed, the location of the incision largely affects the further course of the surgery, and potentially influence both operators’ comfort, but also the risk of periprocedural complications. However, in this regard, the location of the incision does not reflect the incision of the uterus, but of the skin. This information reflects both the location of the placenta and operator preference. Nonetheless, as we do agree that such presentation could mislead the readers, we have added the word “abdominal” in the column title, which now stands as “CS abdominal incision”

I suggest in order to support your conclusion to compare the method parameters (blood loss, transfusion etc) with a hystorical similar lot.

We would like to thank you for this comment. Without doubt, the direct comparison between the historically presented methods and Foley catheter as a tourniquette would be of significance, although we do believe that generation of ideally identical conditions, providing sufficient evidence for comparability of the results would not be possible. 

First, the studied population consists solely of white patients, while other studies evaluating the alternative methods enrolled other races. Moreover, other hardly negligible factors influencing the outcomes, which are difficult to be included in any statistical model, are operators’ experience, as well as the number of operators, local periprocedural management (e.g. the approach to blood transfusion thresholds). 

Therefore, we believe that a generation of a direct comparison between methods could potentially contain a substantial bias, which probably explains why no meta-analyses on this particular subject have been generated so far.

Once again, thank you for your thoughtful feedback. We will make the necessary revisions to address all the concerns you've raised. We hope that our revised manuscript will meet the standards of the journal and contribute meaningfully to the scientific community.

Best regards,

Jakub Staniczek

Reviewer 3 Report

Dear authors,

congratulations

your paper is absolutely of interest, placenta accreta unfortunately is going to be everyday more frequent and your idea regarding the use of a foley catheter is i think really simple and applicable

introduction nice

methods fine

results well presented

conclusions fine

beautiful immages

i would like just to suggest some minor revisions

May you consider to better highlight in general the difficulties of a caesarean section in case of placenta accreta with peripartum hysterectomy please read and cite (PMID: 31962259)

Author Response

Dear Reviewer,

Thank you for your constructive feedback and positive remarks on our paper. We are delighted to hear that our work on the use of a Foley catheter in cases of placenta accreta has been well received.

We appreciate your suggestion to further emphasize the challenges associated with a caesarean section in the case of placenta accreta, particularly when a peripartum hysterectomy is involved. We agree that this is an important aspect that warrants more detailed discussion in our paper. We will revise the manuscript to better highlight these difficulties and their implications.

Regarding the citation you suggested (PMID: 3196225), we will certainly read this paper and incorporate relevant insights into our revised manuscript. We understand the importance of acknowledging and building upon the existing body of knowledge in our field.

Once again, we thank you for your valuable input. We believe that these revisions will enhance the quality and impact of our paper.

Best regards,

Jakub Staniczek

Round 2

Reviewer 1 Report

Dear Dr. Staniczek,

thanks for your detailed and comprehensive response.

Your comprehensive changes improved the manuscript eminently.

I still think that the paper is partly redundant (double information in tables and text, figure 3).

Inspite of your explanation I still do not understand why all these patients got a hysterectomy - with such a low blood loss and only low grade PAS (accrete, increte) in most of the cases and despite the use of a foley catheter.

Research concerning the method's safety and effectiveness in a randomized setting with a higher number of cases is surely necessary.

Presenting relevant results of a new method to handle a difficult situation in obstetrics I would recommend publication in the present form.

Reviewer 2 Report

Thank you for the revisions now the method is comprehensible. 

Qulity of the Enghlish was impoved comparing to the previous version